# Functionality Investigations of Dry-Lubricated Molybdenum Trioxide Cylindrical Roller Thrust Bearings

Dennis Konopka [1,*] , Florian Pape [1] , Norman Heimes [2] , Bernd-Arno Behrens [2] , Kai Möhwald [3] and Gerhard Poll [1,*]

1 Institute of Machine Design and Tribology, Leibniz Universität Hannover, An der Universität 1, 30823 Garbsen, Germany; pape@imkt.uni-hannover.de

2 Institute of Forming Technology and Machines, Leibniz Universität Hannover, An der Universität 2, 30823 Garbsen, Germany; heimes@ifum.uni-hannover.de (N.H.); behrens@ifum.uni-hannover.de (B.-A.B.)

3 Institute of Materials Science, Leibniz Universität Hannover, An der Universität 2, 30823 Garbsen, Germany; moehwald@iw.uni-hannover.de

* Correspondence: konopka@imkt.uni-hannover.de (D.K.); poll@imkt.uni-hannover.de (G.P.); Tel.: +49-511-762-13393 (D.K.)

**Abstract:** In addition to using conventional lubricants, such as oil and grease, rolling bearings can also be used with a dry lubricant. For example, the use of dry lubricant systems is necessary when the application of oils or greases is not possible (e.g., at high temperatures or in aerospace applications). The requirements of a solid lubricant are to reduce friction and wear of mechanical contact partners. In this work, a molybdenum-based coating system was applied by means of physical vapor deposition (PVD). The coating system consists of a molybdenum (Mo) reservoir with molybdenum trioxide ($MoO_3$) as the top layer. The $MoO_3$, which is particularly important for the run-in and the lubricating effect, is intended to continuously regenerate from the reservoir via tribo-oxidation. To determine the friction and wear behavior, cylindrical roller thrust bearings were used. Experiments demonstrated that the lubrication system is effective and that the frictional behavior has been improved. On the one hand, the frictional torque of the rolling bearings has been considerably reduced and, on the other, significantly extended operating times have been determined compared to unlubricated reference experiments. Simultaneously, material analyses have been carried out by means of scanning electron microscopy (SEM) and energy dispersive X-ray spectroscopy (EDX). The investigations showed that the $MoO_3$ was transferred to uncoated bearing components. This improved the tribological behavior and reduced abrasive and adhesive wear.

**Keywords:** rolling bearing; dry lubricant; Mo-coating; PVD; friction; wear



## 1. Introduction

From the perspective of climate change and the increasing consumption of fossil elements, systems with frictional losses are constantly being optimized in order to become more efficient. In addition, to help reduce the energy consumption during operation, the life cycle costs of a system increasingly come into focus. Rolling bearings are used in almost every machine with rotational movement. Therefore, design changes have been made in recent years to optimize friction and wear properties [1]. A general econometric review with classification of significance of tribology was published in 1966 [2] and updated in another review by Jost in 1990 [3]. The author shows that, due to the technological rapid developments, unanswered questions can only be solved by the interaction of tribology and materials science. For this reason, in addition to design changes, lubricants are also being further developed with regard to new requirements [4]. In this paper, the focus is on a dry lubricant for rolling bearings in order to enable improvements for conditions that cannot be covered by classic oil or grease lubrication. These conditions and scenarios include, for example, extreme temperatures, in the food industry, or space technology. In

order to enable the preparation of such solid lubricants, there are several concepts and manufacturing processes. Various pure elements or composite substances are used. An overview of common solid lubricants is given by Birkhofer and Kümmerle [5]. Generally, it is noted that the use of dry lubricants is application-specific and works for a certain or very limited field of operation. In current research, molybdenum disulfide ($MoS_2$) is often the subject of investigation as a solid lubricant. A review with a focus on $MoS_2$ is given by Vazirisereshk et al. [6].

For this work, a molybdenum-based coating system was prepared by means of physical vapor deposition (PVD). For this purpose, thin molybdenum-based layers were deposited on the substrates. The coating system consists of a dominant molybdenum (Mo) reservoir and a lubricating top layer of molybdenum trioxide ($MoO_3$). The Mo, as well as the $MoO_3$, are deposited with a molybdenum target [7]. It is of particular interest in tribology to develop a dry lubricant for solid-state contacts between two metallic objects that can lubricate in the case of direct mechanical contact but also provides wear protection. Therefore, different coating systems were developed in order to provide $MoO_3$ in the tribological run-in phase. The coating systems were deposited on bearing steel AISI 52100. In the studies by Schöler et al., the layer thickness of the top layer was 500 nm. The reservoir, which provided a continuous supply of lubricant in the tribocontact by tribo-oxidation, had a layer thickness of 2 μm [8]. The newly developed coating systems have been fully analyzed to obtain their mechanical properties. In order to investigate the lubricating mechanisms of the dry lubricant in more detail, oscillating sliding tests were carried out in the microscale on laboratory substrates by Konopka et al. The result was that the first deposited molybdenum-based coatings delaminated and did not have enough adhesion to the substrate [9]. Reducing the coating thickness of the top layer gave a better initial point for the tribological contact. The reason for this was the lower residual stresses of the coating system. Furthermore, the wear particles were finely dispersed and improved the friction behavior. It was also determined that a dry lubricating layer was formed and a transfer of Mo particles to the counterbody in the sliding contact occurred. To avoid delamination due to increased stresses, the coating thicknesses have to be made especially thin. Therefore, additional material characterizations and indentations were performed on the nanoscale. Wear characterization at the microscale was not possible because of the thin molybdenum oxide coatings' high wear rate. Consequently, the wear behavior of the Mo and $MoO_3$ coatings was characterized and modeled by nano-wear tests [10]. It was shown that $MoO_3$, due to its low hardness compared to the harder Mo reservoir, is distributed particularly quickly with a lubricating effect in the tribological contact. Different approaches to the transfer of tribological effects between nano- and microscales have been investigated [11].

The $MoO_3$ coating systems developed need to be applied to commercially available machine elements and analyzed from a tribological point of view. The main focus in this work is the usability and efficiency of the developed coating system in rolling bearing applications. In addition, the tribological investigations determined in the laboratory are to be extended to real components like rolling bearings. There are various types of bearings that can be considered for the investigations. Heinz-Schwarzmaier investigated the operating life of commercial ball bearings (type 6001) under extreme conditions. The bearing rings were heated to a temperature of around 300 °C and tested in a vacuum. A molybdenum disulfide and silver coating was used. The dry lubricant itself was only deposited on the rolling elements [12]. Further approaches to determine the operating life are provided by test rig investigations of Pörsch et al., who analyzed dry-lubricated angular contact ball bearings of type 7205 in a four-bearing test rig [13]. The bearing ring surfaces were coated with $MoS_2$. At the same time, the cage pockets were modified and used with a friction-reducing compound of tungsten disulfide ($WS_2$). The focus of the investigations, in addition to the determination of the operating life, was the lubricant transfer mechanism [13] and the release of dry lubricant particles by the rolling cage itself. Advanced wear analysis of the experiments mentioned were extended and validated with simulations by Dahiwal et al. [14,15]. In addition to lubricant transfer in dry-lubricated rolling bearings,

the forming of tribofilms is also the subject of investigation. Typically, tribofilms may form during operation of the thrust bearings, which can prevent abrasion [16]. Pape et al. investigated the behavior of graphene platelets as a solid lubricant on angular contact ball bearings. Under oscillating operating conditions, an advantageous low frictional torque was measured and a lower level of wear was even detected compared with conventional lubrication [17,18].

The aim of these investigations was to verify the functionality of the developed Mo-based system and to determine operating life. For this purpose, a cylindrical roller thrust bearing of type 81212 was used. The following questions should be answered by the investigations:

- Does the developed coating system have a positive effect and does it prevent tribological wear?
- Is it possible that the molybdenum trioxide improves the friction behavior of the chosen rolling bearing and reduces the frictional torque?
- Is there an interaction or transfer of the $MoO_3$ lubricant to uncoated bearing components?

## 2. Materials and Methods

### 2.1. Sample Preparation

Before presenting the experimental methods and the performance of the tests, the coating processes and specimens used will be described in the following. As mentioned before, a PVD process was used to coat a bearing with molybdenum and molybdenum trioxide. PVD is a vacuum-based deposition process that enables the application of hard, thin coatings on substrates to specifications. For this study, the molybdenum-based coatings were created by magnetron sputtering. A CC800/9 (CemeCon, Wuerselen, Germany) system was chosen. Pure Mo sputtering targets (with purities of 99.95%) were used for coating. Figure 1 shows the general tribological system of a rolling bearing. Figure 1a shows the components of the bearing. The bearing consists of two bearing washers, a rolling element cage, and cylindrical rollers. All components, except for the cage, are made of annealed 1.3505 (AISI 52100, 100Cr6), a hardened bearing steel (see Table 1).

**Table 1.** Chemical composition of used bearing steel in % mass fraction.

| C | Si | Mn | P | S | Cr | Mo | Al | Cu | O |
|---|---|---|---|---|---|---|---|---|---|
| 0.93–1.05 | 0.15–0.35 | 0.25–0.45 | 0.025 | 0.015 | 1.35–1.60 | 0.10 | 0.05 | 0.30 | 0.0015 |

The cage, which carries the rolling elements, is made of a PA66 material. In this work, the thrust washers (see Figure 1b,c) were coated to study the behavior of the new friction-reducing solid lubricant. In the following, the PVD process parameters and manufacturing steps will be explained. Before the rolling bearing washers are inserted into the PVD system for coating, cleaning steps are necessary. For this purpose, the whole surface was put in an ultrasonic bath, first using isopropanol and then acetone, each for 15 min, to remove oily conservatives. After the insertion of the specimens into the PVD system, air was pumped out of the recipient until a pressure of $6 \times 10^{-5}$ Pa (i.e., below atmospheric pressure) was reached.

Due to the reaction-affinity of oxygen with the bearing steel substrate, thin oxidation layers can be formed. This negatively affects the adhesion and bonding of the dry lubricant. For this reason, the surface of the substrate was cleaned for 5 min by material-removing plasma etching before sputtering, and this was done while the vacuum was applied. The steel substrate was heated to a steady-state temperature of 200 °C. Tempering effects can be neglected at this stage. Next, the sputter (vacuum) chamber was filled with argon at a flow rate of 50 sccm and a volume pressure of $350 \times 10^{-3}$ Pa. The cathode power was set to 4000 W and the BIAS voltage was 100 V. The distance from the Mo target to the substrate was 100 mm. The sputtering power and duration were selected for the pure

molybdenum to achieve a layer thickness of 2 μm. For the preparation of the top layer, the thrust bearing washers were placed in a continuous furnace for 40 min after sputtering. The furnace has a constant temperature of 300 °C. This process step causes oxidation of the molybdenum surface. This produces a $MoO_3$ surface with a thickness of approximately 100 nm as a toplayer.

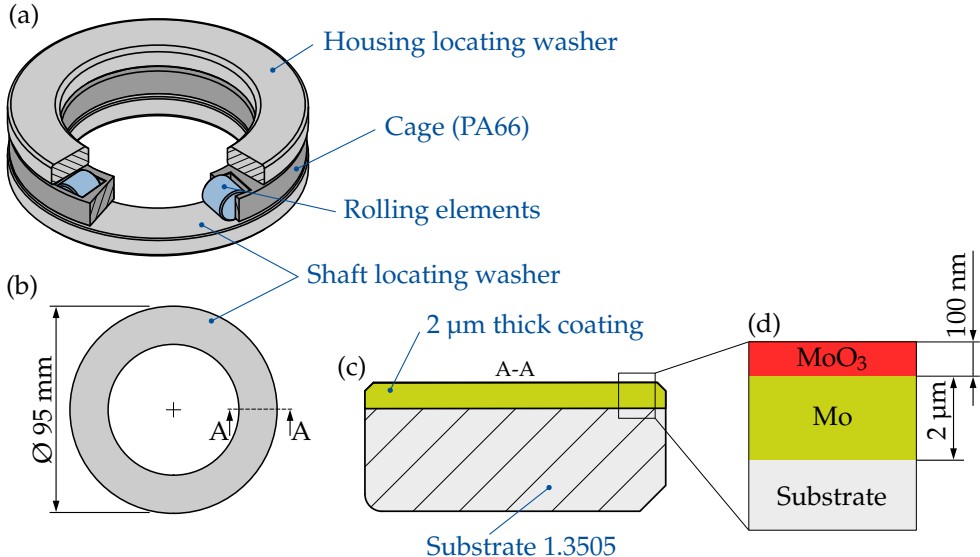

**Figure 1.** Overview of the prepared specimen. (**a**) Cylindrical roller thrust bearing 81212. (**b**) Top view of the coated bearing washer. (**c**,**d**) Cross-sectional view of the coating system (not to scale).

The roughnesses of the bearing washers were determined before and after the coating process. The layer deposition in the PVD process resulted in a minor change in roughness. This roughness change is common. Furthermore, the roughness changes due to the coating were still within the manufacturing-generated roughness tolerances of the bearing washers. Before coating, the roughness was $Ra = (0.116 \pm 0.007)$ μm. Afterwards, a roughness of $Ra = (0.113 \pm 0.011)$ μm was determined. The measurements were repeated at 10 different positions by using 3D laser scanning microscopy.

In addition to the deposition, the layer systems were characterized with a nanoindenter. The nano hardness and modulus of elasticity of the Mo reservoir and the $MoO_3$ system were determined according to the method of Oliver and Pharr [19]. A Hysitron TriboIndenter TI 950 (Bruker, Minneapolis, MN, USA) with a triangular Berkovich diamond tip with a tip radius smaller than 50 nm was used. The Berkovich tip was calibrated on a fused quartz sample. A minimum penetration depth of 28 nm was determined, which must be exceeded during indentation. The layer thickness for the Mo reservoir was set to about 2 μm (see Figure 1d), so that a normal force of 5000 μN could be selected, since the minimum penetration depth was exceeded and the 10 % rule was observed [20]. For the Young's modulus of the Mo reservoir, an elasticity of $(224.92 \pm 10.62)$ GPa was measured; the nano hardness was $(6.18 \pm 0.78)$ GPa. Due to the lower layer thickness of the $MoO_3$ layer system, the normal force was reduced to 500 μN, since the minimum penetration depth of 28 nm could just be exceeded there. The modulus of elasticity of the top layer was $(122.62 \pm 14.85)$ GPa and the nano hardness was $(2.45 \pm 0.48)$ GPa.

## 2.2. Experimental Setup

In order to verify the functionality of the molybdenum-based system, the coated specimens were subjected to tribological stress in a modified FE8 test rig. Different types of rolling bearings can be used in the test rig. A test head for cylindrical roller thrust bearings designed by the IMKT Hanover was used for these investigations. According to

the standard *DIN 51819* [21], the test rig was designed for fatigue and wear tests on rolling bearings using automotive and industrial lubricants such as oil and grease. However, the investigations in this paper are focused mainly on the lubrication mechanism of a solid lubricant. In the FE8 test head, two thrust bearings of type 81212 were mounted on one shaft, see Figure 2. Because of the geometry of cylindrical roller thrust bearings, a pure rolling motion will only occur in the center of the rolling element. In the other areas further outside and inside, slip between the rolling elements and the housing or shaft located washer will occur (Figure 2b). This relative movement increases towards the rolling element ends and reaches a slip level of up to 14%, depending on the operating conditions.

For the bearing type 81212, the outer diameter is Ø 95 mm. For the inner hole, a diameter of Ø 60 mm for the the shaft washer (WS), and an inner hole of Ø 62 mm diameter for the housing washer (GS) were given. After the manufacturing process, roughness and surface profiles were measured by using a 3D laser scanning microscope.

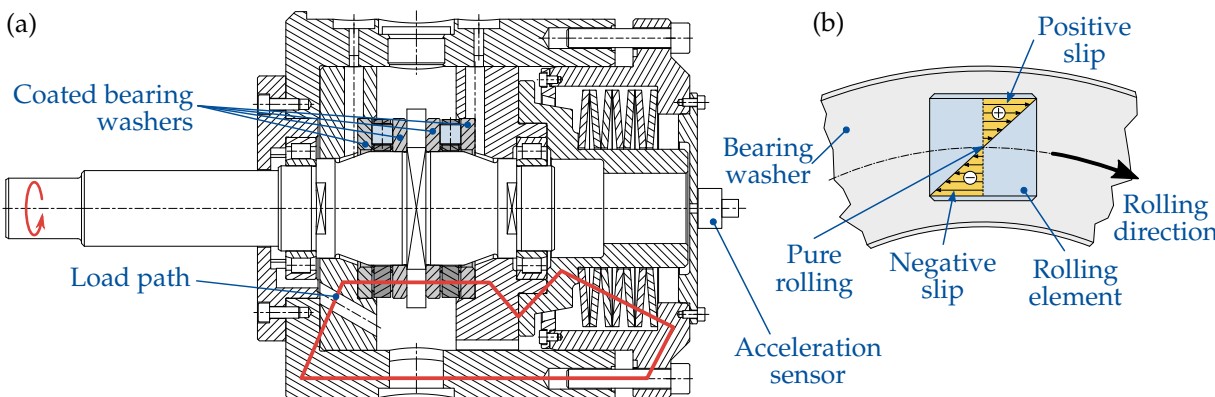

**Figure 2.** Schematic representation of the test rig and the tribological system: (**a**) FE8 test head with type 81212 cylindrical roller thrust bearing. (**b**) Slip distribution in axial cylindrical roller thrust bearings displayed on the bearing washer.

The drive engine is connected to the test unit by a non-contact torque sensor of type 8661 (Burster Praezisionsmesstechnik, Gernsbach, Germany), that contains an incremental encoder to allow angle-synchronized torque measurement. The axial force was applied by a disc spring assembly. In addition to the frictional torque measurement, PT100 temperature sensors were used to measure the temperature of all four bearing washers and acceleration sensors KS80D (MMF, Radebeul, Germany), which constantly monitor the vibration signal of the system. The sensor is mounted centrally on the front of the load unit (see Figure 2a, right) of the test head.

### 2.3. Experimental Conditions

For the following investigations, the test bearings were loaded with an axial force of 15 kN. A maximum contact pressure of 1 GPa results from the evenly distributed force. The experiment was conducted under steady-state ambient conditions with an atmospheric temperature of approximately 21.5 °C at the test rig and a rotational speed of 50 rpm. An overview of the test conditions is shown in Table 2.

**Table 2.** Experimental conditions.

| Parameter | Value | Unit |
|---|---|---|
| Axial load $F_{ax}$ | 15 | kN |
| Speed $n$ | 50 | rpm |
| HERTZian contact $p_{H,max}$ | 1 | GPa |
| Number of rolling elements | 15 | – |
| Ambient temperature $T_{amb}$ | 21.5 ± 2.0 | °C |
| Relative humidity $\varphi_{amb}$ | 55 ± 10 | % |

As a first step, uncoated bearing components were investigated as a reference. The aim was to determine the operating life of standard bearing systems under dry-running conditions and compare these results with the developed Mo-based coating system. The operating life is defined as the running time until a cut-off criterion has been reached. The cut-off criteria were exceeding a temperature limit, torque limit, or a vibrational threshold. The test bearings with the coating system, as well as the reference bearings without coating, are mounted in the test head in dry conditions. The surfaces are therefore free of oil and grease in the initial state.

### 2.4. SEM, EDX, and Optical Investigations

For the material-specific investigation of the coating system, high-resolution analytical methods were used. These analysis methods allow the determination of element concentrations, mass occupancy, and residual coating thickness. The wear tracks were analyzed by scanning electron microscopy (SEM) and energy dispersive X-ray (EDX) spectroscopy using a Supra 40VP (Zeiss, Oberkochen, Germany). The focus was to investigate the wear tracks after the experiment, as well as the reactions of the respective surfaces of the tribological counter bodies. Furthermore, the worn surfaces were analyzed by EDX mappings. For measurement of roughness and characterization of surfaces before and after the experiment, a VK-X100K 3D laser scanning microscope (Keyence Corp., Osaka, Japan) was used. A laser sensor can be used to scan the state of the surface without mechanical contact and record the coating system, the wear track, and detached particles after the experiment. This enabled a high-resolution damage analysis. The surfaces of the contacting bodies were scanned separately. This allows a detailed investigation of detached particles after the test.

## 3. Results

### 3.1. Determination of a Cut-Off Criterion

In the following section, the performed tests will be presented and compared with each other. The first part of this section discusses the definition of a cut-off criterion for the dry-running thrust bearings under a contact pressure of $p_H = 1\,\text{GPa}$ and a speed of $n = 50\,\text{rpm}$. In addition to the temperature development, the investigation also focused on the vibration signal during the test. However, primarily the frictional torque and the behavior of the drive machine were observed and monitored. The results of the reference (Ref. 1 and Ref. 2) tests are shown in Figure 3. The experiment was repeated under the same conditions to verify reproducibility.

The investigations showed that, after a short running-in phase, there was a significant increase in frictional torque, see Figure 3. The acoustic effect of the experiment was characterized by an unpleasant squeaking noise from the test bearings. A constant frictional torque was determined after a test time of about 5 min in both reference tests. The frictional torque for Ref. 1 and Ref. 2 was approximately 39 Nm. This phase could be observed for another 5 min until a linear increase of the signal occurred. This linear behavior occurs more intensely with Ref. 1. The experiment was stopped after a test duration of 20 min and a torque of 52.5 Nm; otherwise, mechanical damage to the drive or the torque transducer could result. In the second reference test, a linear increase of the friction torque up to a value of 52.9 Nm was also observed. The test duration was 38 min. On a more detailed analysis in terms of time, it can be seen that the running-in phases of the two non-lubricated thrust bearings were comparable and started at a frictional torque of 6 Nm. Qualitatively, it can be noted that a first maximum of the frictional torque was recorded after 2 min.

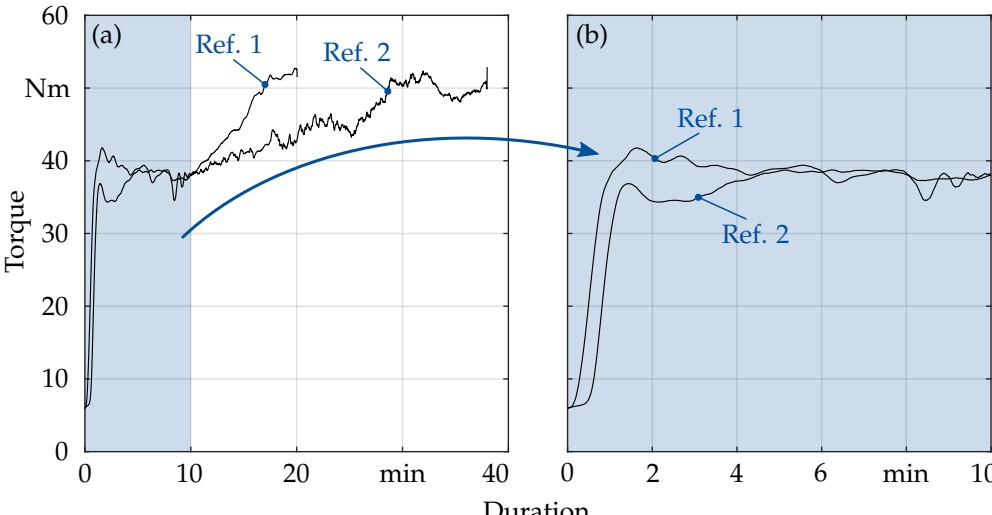

**Figure 3.** Overview of reference tests Ref. 1 and Ref. 2: (**a**) Frictional torque of the reference experiments to determine the operating life without lubricant displayed over the operating time. (**b**) Frictional torque curves observed in more detail from (**a**). Shown are the first 10 min with focus on the running-in behavior.

After the reference tests, the bearing washers were dismounted from the test head and prepared for optical microscopy. In the following, the images of the bearing from reference experiment Ref. 2 will be presented. The focus of the investigations was on the area of worn rolling contact material. The worn raceway is shown in Figure 4a,b. Qualitatively, it can be noted that a significant change of the surface has occurred over the entire wear track. For comparison, Figure 4c shows an image of the initial state from the reference surface. Furthermore, the colored (reddish) changes that are more concentrated in the center of the wear track are clearly visible. This colored modification indicates tribo-oxidation. This occurs particularly strongly in HERTZ'ian contacts with dry sliding and rolling friction. Figure 4e shows this area in more detail. On the one hand, it can be seen that material has been worn abrasively and a tribological reaction has occurred. Another interesting area is shown in Figure 4d. At this position, increased sliding due to the particular kinematic situation in the thrust cylindrical roller bearing occurs, meaning that the sliding effect has an increased influence. A clear change of the surface is the result of this. In general, this area of the wear track was polished by the experiment. This was also verified by the measurement of roughness parameters. After the test, at the position with the highest slip (see Figure 4d), a roughness decrease to $Ra = 0.046\,\mu m$ was measured.

This is a typical result of a relative movement between a rolling element and the bearing washer in solid contact. It can be assumed that the wear behavior is most significant at this position. The formation of a tribo-oxidation film has not been detected at this position. After the test, it was noted that the surfaces of the bearing washers show a clear topographical and mechanical change after a test duration of 38 min and that the operating life under these conditions had been reached. The critical factor was the increase of frictional torque and the unstable movement of the rolling bearings. When considering the measured frictional torque, it was possible to define a cut-off criterion of 45 Nm by means of the reference tests. This value was used as a cut-off criterion for the ongoing investigations of the molybdenum-based coatings.

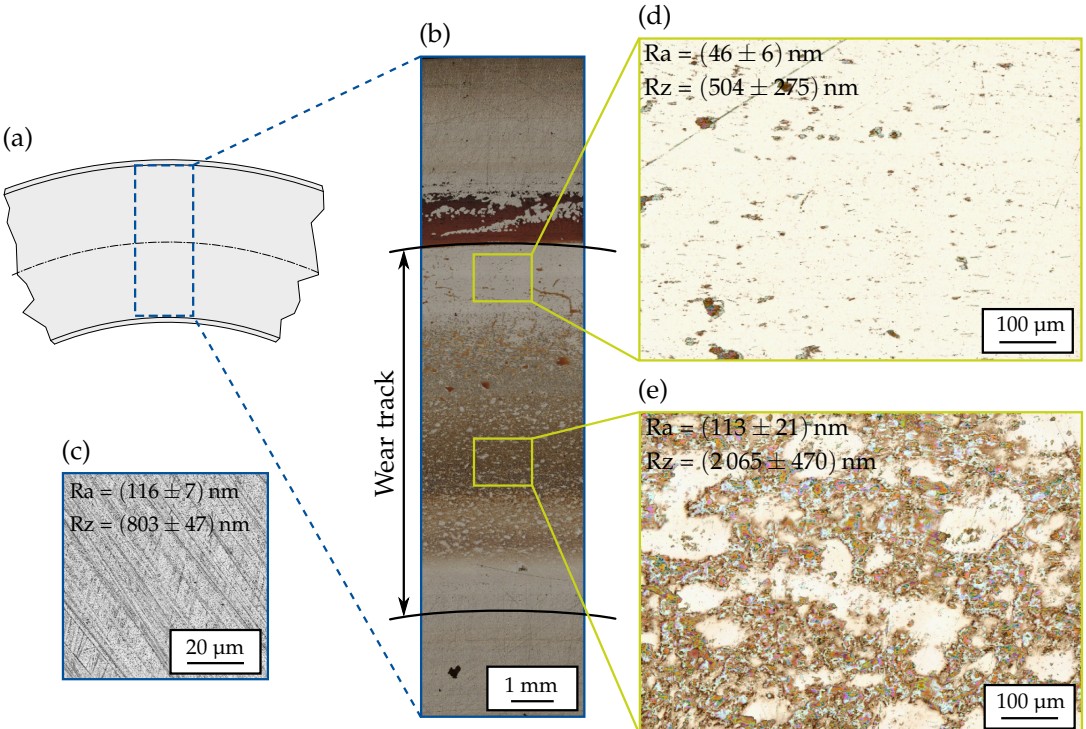

**Figure 4.** Damage analysis of the reference specimen Ref. 2 by 3D laser scanning microscope: (**a**) Principal section view of a bearing washer. (**b**) Wear track of the reference specimen after a test duration of 38 min. (**c**) Bearing washer in initial state with measured roughness parameters *Ra* and *Rz*. (**d**,**e**) High magnification microscopic view of the wear area with Ra and Rz.

### 3.2. Molybdenum Trioxide Coatings

The first experiments with cylindrical roller thrust bearings without lubricant gave information about the system behavior; the investigation was repeated with the molybdenum trioxide coatings, taking the determined cut-off criterion into account. The operating life is the time until the cut-off criterion (max. frictional torque of 45 Nm) is reached. The time series of the measured torque is shown in Figure 5. The horizontal red line displays the cut-off criterion. During the test, different behavior and interesting operating conditions occurred. Therefore, the tests are presented in three diagrams. On the one hand, this allows an overview of the entire series of tests to be presented in terms of time, and on the other hand, the run-in-specific operating characteristics can be described.

Twelve bearings were available for investigation. Therefore, it was possible to conduct six experiments with the test setup shown in Figure 2. The criterion was reached at different experimental times. Taking a closer look at Figure 5a, it can be seen that there are some experiments that reached the cut-off criterion after a time of around 500 min. In relation to this, there is also an experiment which reached a test duration of up to 2900 min (48 h). In the first 200 min, the experiments had qualitatively comparable characteristics in the measured signal. This can be seen in the more detailed view of the frictional curves in Figure 5b. During the running-in phase of the modified bearing with $MoO_3$, the solid contact resulted in a maximum frictional torque in the first 10 min. Subsequently, it can be seen that, in all tests, the operating behavior recovered tribologically and the frictional torque was reduced again to a value of about 20 Nm. This frictional torque was reached after about 50 min. Afterwards, there was a linear increase in friction. This can be observed in all tests. However, the gradient varied to different degrees depending on the test and has a time dependent influence on reaching the cut-off criterion and thus on the operating life of the respective system. For a closer look into the run-in behavior of the coated systems, the frictional torque plot in the first 40 min is summarized in Figure 5c. To show

the performance of the coating in comparison, the uncoated reference experiments are featured in the same diagram. It can be seen that the molybdenum-based coating systems (blue curves) had a frictional torque of 6 Nm initially. The same behavior was also observed in the reference tests (black curves). However, the determined maximum frictional torque of the reference test was reached much earlier in the direct time comparison. Furthermore, a significant reduction of the frictional torque can be seen, which indicates the lubricating effectiveness of the molybdenum trioxide. An overview of the experiments is shown in Table 3.

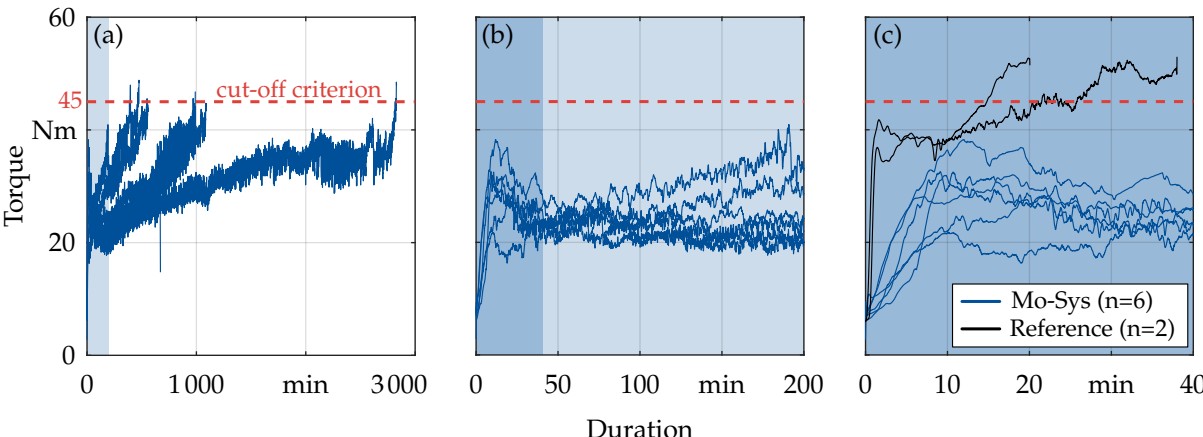

**Figure 5.** Frictional torque curves of the tribological investigations of the molybdenum-based coating systems: (**a**) Frictional torque development of the experiments as an overview (*n* = 6) over time. (**b**) Higher time resolution of the Mo experiments. Detailed diagram of blue area from (**a**). (**c**) Time trend of the tests limited to the first 40 min and a comparison of the molybdenum system and the reference tests.

**Table 3.** Experimental overview and comparison of the *Mo-based* and *Reference* specimens.

| | | **Mo-Based System** | | | | | | **Reference (Uncoated)** | |
|---|---|---|---|---|---|---|---|---|---|
| | Specimen | 1 | 2 | 3 | 4 | 5 | 6 | Ref. 1 | Ref. 2 |
| $T_{10}$ [1] | Nm | 15.3 | 22.1 | 16.3 | 20.1 | 23.9 | 16.2 | 34.6 | 36.8 |
| | mean | | | $18.97 \pm 3.26$ | | | | $35.71 \pm 1.13$ | |
| $T_{40}$ [2] | Nm | 18.6 | 26.2 | 22.6 | 24.9 | 30.3 | 23.3 | 40.3 | 43.2 |
| | mean | | | $24.32 \pm 3.57$ | | | | $41.70 \pm 1.43$ | |
| $t_{cut-off}$ [3] | Minutes | 671 | 476 | 559 | 2827 | 990 | 1092 | 14 | 16 |
| | mean | | | $1102.48 \pm 802.24$ | | | | $15.10 \pm 1.11$ | |

[1,2] Time-averaged torque $T$ as a function of a experimental time of 10 min and 40 min in Nm. [3] Duration $t$ until cut-off limit (45 Nm) was reached in min.

### 3.3. Evaluation of the Rolling Bearings after the Experiment

After a bearing reached the cut-off criterion and thus the operating time was reached, the bearing components were investigated. In addition to the general condition after an experiment, the tribologically stressed area of the contact between the rolling bearing washer and rolling element was of particular interest. Besides, the research issue of whether the $MoO_3$ works as a lubricant in solid contact and prevents abrasive and adhesive wear, a possible regeneration by tribochemical reactions was the subject of investigation. For this reason, the surface of the coated bearing washers was analyzed with regard to wear, but also with regard to the materials and boundary layers formed. For this purpose, high-resolution material investigations were performed using SEM and EDX. For the following damage analysis, the coated system with the longest test duration (48 h) was used. In order to give



an overview of the wear behavior, the bearing washer is shown in Figure 6. Although the frictional torque maximum of 45 Nm was reached, the surface of the coated system still looked intact. In contrast, the uncoated reference tests showed clear traces of abrasive wear and adhesion, see Figure 6d,e.

In direct comparison between the wear track (Figure 6b) and the surface in its initial state before the test (Figure 6c), the wear track formed by the rolling contact was clearly visible, and different colors appeared on the surface when viewed vertically. This can also be shown again in a much more differentiated way in Figure 6d,e. Furthermore, the production-specific roughness and grinding patterns are also no longer visible. However, it is noticeable that there was no metallic surface spalling due to the solid contact. It can be clearly seen that a tribochemical reaction took place in the area of the highest slide to roll ratio. As a reminder, this area is at the two ends of a roller element in tribological contact of a cylindrical roller thrust bearing. Especially in these areas, it seems that a formation of reaction films or boundary layers occurred. To get further information about the reaction film, near-surface analyses were performed using SEM, see Figure 7. For the examination of the bearing washer, an area was selected where a certain proportion of a non-worn as well as worn surface could be detected and analyzed. This allows a comparison and shows a change due to wear on the surface. This area is shown in Figure 7a. This figure shows the macroscopic surface of a thrust bearing washer used.

In Figure 7b an SE image is shown for an overview of the generated EDX mapping. This makes it possible to visualize the remaining wear particles after the test and to visualize the worn surface sufficiently. The separation of the different elements itself is shown in Figure 7c by the EDX composite spectrum.

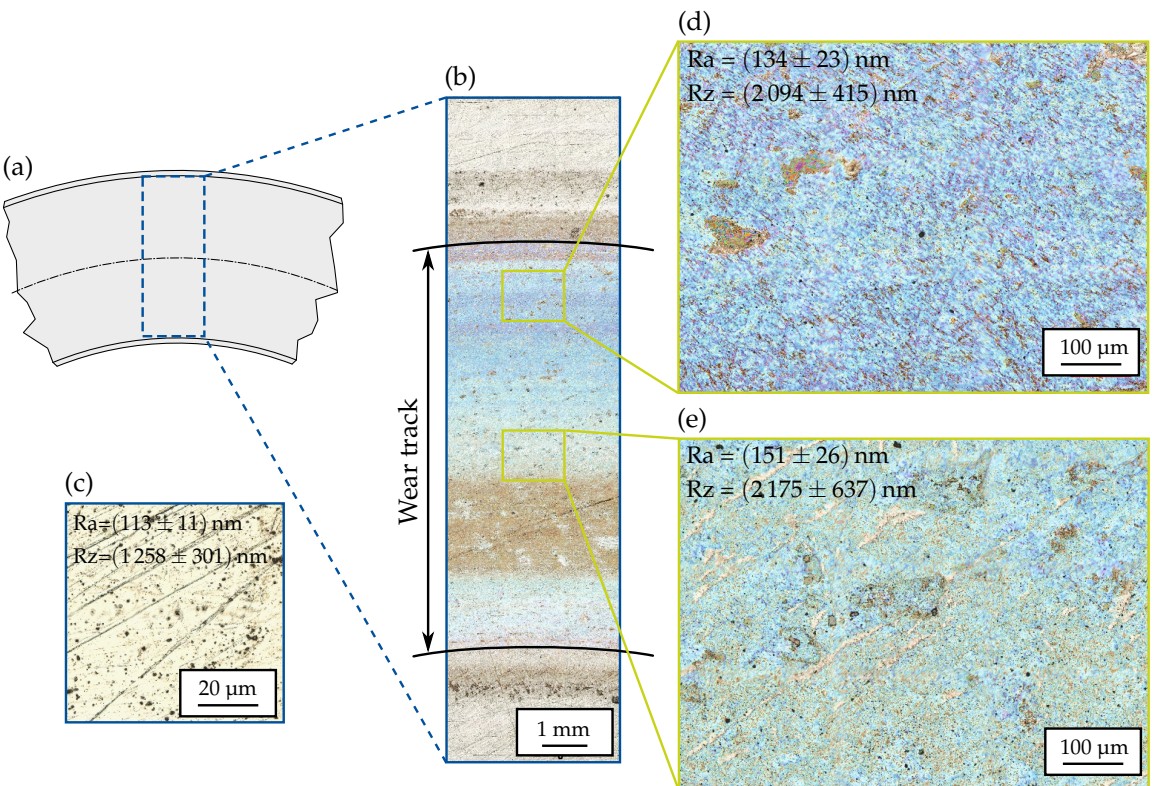

**Figure 6.** Damage analysis of a $MoO_3$ specimen by 3D laser scanning microscope: (**a**) Principal section view of a bearing washer (schematic). (**b**) Overview of the wear track after a test duration of 48 h. (**c**) Bearing washer in initial state with measured roughness parameters Ra and Rz. (**d**,**e**) High resolution microscopic view of the wear area with Ra and Rz.

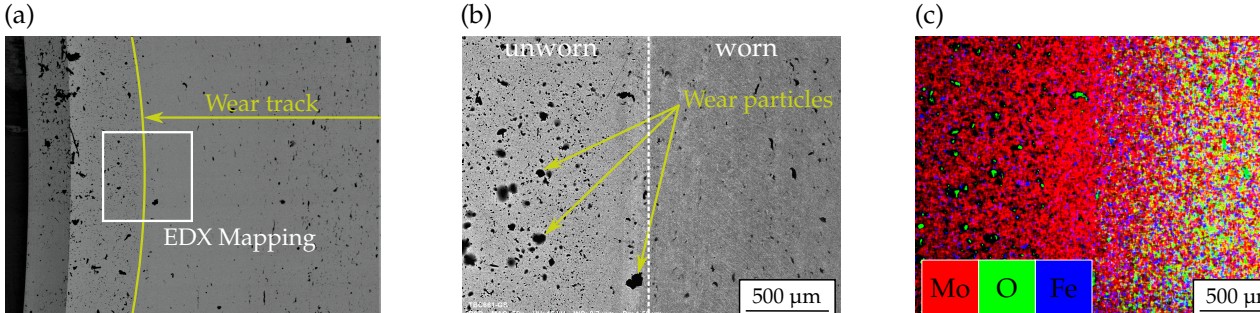

**Figure 7.** EDX investigations after the experiment on bearing washers: (**a**) Extended view of the bearing washer after experiment with defined area for EDX mapping. (**b**) SE image using InLens detector of the EDX mapping area with unworn and worn area. (**c**) Qualitative elemental mapping of *Mo*, *O*, and *Fe* from the inspected area in (**b**).

The EDX analysis shows that Mo is still present in the wear track after the experiment. There is also an increased oxygen concentration, which is attributable to tribo-oxidation from the Mo reservoir. The wear-reducing properties of the Mo could therefore be effective, as in addition to the minimal abrasive wear, no cold welding could be detected on the raceway. The observed wear particles themselves can be explained partially by the wear of the coating. Furthermore, after the experiments, it was determined that the rolling element cages inside the cylinder pockets were slightly worn. The cage itself is made of a polymer material (*PA*66) and has the function of supporting the rotational movement of the rolling elements, keeping them spaced evenly, and causing an even load distribution in the rolling bearing. However, the cages themselves are not the focus within this work. The worn pockets showed that, the reference tests in particular (i.e., the specimens without coating), showed a higher degree of dry-contact wear than the molybdenum-based coating systems. The cylindrical rollers themselves initiate the wear of the cage or cage pockets. Therefore, the surface topographies of the rolling elements were examined and investigated in more detail. In addition, the question arises as to whether the molybdenum or molybdenum trioxide has a positive influence on the wear and friction behavior of the uncoated cylinder. It should be mentioned that only the bearing washers have been coated with the molybdenum trioxide. The rolling elements themselves were unlubricated for the investigations. The cylindrical rolling elements used are made of rolling bearing steel 1.3505 (*AISI* 52100). The following Figure 8 shows the surface of the rolling elements.

A clear change in the rolling element surface can be seen. Figure 8a shows a rolling element from the reference test. A relatively inhomogeneous reddish change is obvious, which indicates a high affinity of tribo-oxidation. In contrast, no cold welds or any corrosion products like rust were detected on the rollers of the molybdenum-based specimens (see Figure 8b). In general, it can be observed that the investigated surfaces of the rolling elements were in a good condition after the test with an extended test duration. Furthermore, the color change detected on the bearing washers (Figure 6d) within the slip zones can be seen on the counterbody.

Because material-specific investigations by means of an EDX demonstrated that molybdenum residues could be detected on the rolling bearing washer in these areas, these investigations were repeated on the rolling element surfaces. A tribochemical reaction layer was detected on the surface. The main question was whether the molybdenum would transfer to the rolling element surface by tribological transfer. In essence, the tribochemical modification can be detected on the whole cylinder and also the EDX investigations showed that a fraction of molybdenum is present (see Figure 8c).

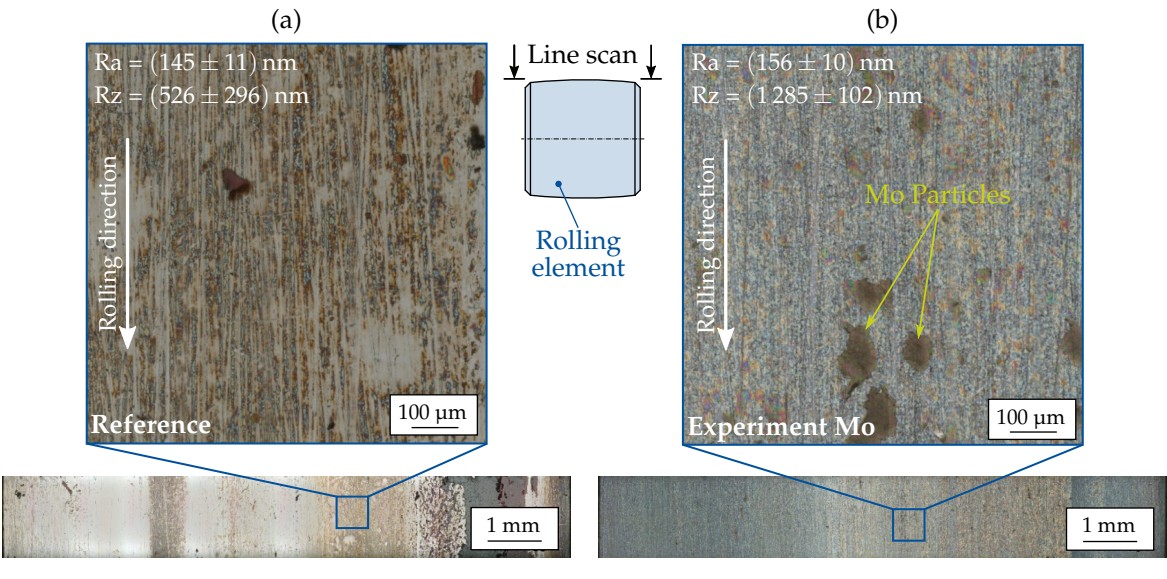

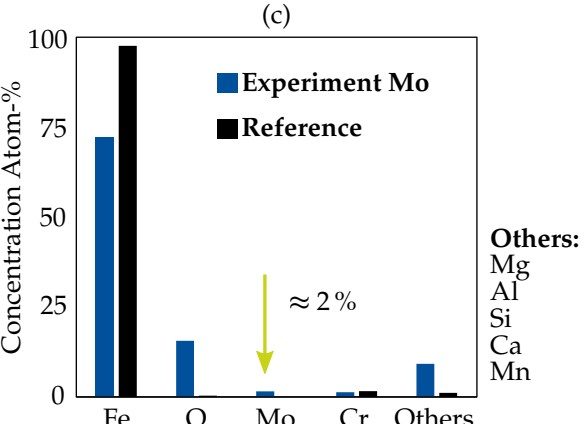

**Figure 8.** Surface analysis of the rolling elements: (**a**) Representation of the reference experiment. Significant accumulation of oxides due to tribo-oxidation and partial cold welding could be observed. (**b**) Mo-based system induced a homogeneous change in the color of the surface. (**c**) EDX analysis of the *Mo-based* and the *Reference* experiment after the tests.

## 4. Conclusions

In this work, a newly developed coating system consisting of the combination of molybdenum and molybdenum trioxide was investigated in rolling element bearings under high mechanical loads. The purpose of the investigation was to demonstrate the effect of lubrication and to validate the adhesive and abrasive wear behavior in comparison with uncoated rolling bearings in dry contact conditions.

The results of the research can be summarized as follows:

- The frictional torque of the system compared with the reference tests was significantly lower and was reduced by up to 60% in a direct comparison.
- In the experiments, a significant extension of the operating life was demonstrated for the dry-lubricated rolling bearings compared with unlubricated references.
- Uncoated components like the rolling elements could be protected by the coating system in the experiment and no reactions due to tribo-oxidation could be detected within the wear track.

Further research approaches can be derived from the results. The investigations showed that the experimental duration of the dry-lubricated rolling bearings was extended and the cut-off criterion was reached significantly later compared to the reference tests. This

indicates a longer operating life for the molybdenum-based systems. Statistical evaluation could be used to determine the operating life of the dry-lubricated rolling bearing. This may require further experiments to reach a statistically significant conclusion. Furthermore, methods such as the Weibull test for determining fatigue life of rolling bearings can help statistically to enable prognoses to be made. It was determined that lubricant transfer, i.e., of the dry lubricant to uncoated bearing components, had occurred. This lubricant transfer, together with the additional formation of a tribofilm, will be examined in more detail in future research work. All in all, these results mean that further development steps are necessary. This enables further improvements and also industrial use seems possible.

**Author Contributions:** Conceptualization, B.-A.B., G.P. and K.M.; methodology, D.K.; software, D.K.; validation, D.K.; investigation, D.K.; data curation, D.K.; writing—original draft preparation, D.K.; writing—review and editing, F.P. and N.H.; visualization, D.K.; supervision, G.P. and F.P.; project administration, B.-A.B., G.P. and K.M.; funding acquisition, B.-A.B., G.P. and K.M. All authors have read and agreed to the published version of the manuscript.

**Funding:** This research was funded by the German Research Foundation (Deutsche Forschungsgemeinschaft, DFG), grant number 407673224.

**Institutional Review Board Statement:** Not applicable.

**Informed Consent Statement:** Not applicable.

**Data Availability Statement:** Data is contained within the article.

**Acknowledgments:** The results presented in this paper were obtained within the scope of the priority program "Fluidless Lubrication Systems with high Mechanical Load" (SPP 2074) in project 2, funded by the German Research Foundation (Deutsche Forschungsgemeinschaft, DFG)). Grant number 407673224. The authors gratefully acknowledge the German Research Foundation for their financial support of this project. The publication of this article was funded by the Open Access Fund of the Leibniz Universität Hannover.

**Conflicts of Interest:** The authors declare no conflict of interest.

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
