# Peer review of "Functionality Investigations of Dry-Lubricated Molybdenum Trioxide Cylindrical Roller Thrust Bearings"

_coatings, doi:10.3390/coatings12050591_

Round 1
Reviewer 1 Report
The work deals with the important issue of bearing lubrication. It is interesting but needs improvement. My comments below.
- The title issue - consider changing the title, you haven't determined the bearing life due to the occurrence of seizure or pitting damage.
- Lines 7 and 15. Correct the marking of MoO3 (add subscript)
- Line 94 A very common cause of bearing failure is surface pitting. However, the long-term lubricant was not tested on the test stand until fatigue chipping appeared. Why have not such tests been performed, e.g. in the ball-cone system, which are shorter?
- Referring to Figure 2, explain the positive and negative slippages in the text Line 108 Enter the name of the coating equipment used
- Line 115, if possible, provide the designation of AISI52100 steel in the 1.XXXX format. Give the chemical composition and properties of the steel.
- Line 118 What was the roughness of the steel on which the coating was applied. The result should be given in the system Y = X ± δ. The issue of the reference area should be clearly defined already in Chapter 2.
- Fig 1c Give the symbol of the applied coating with the description "2 μm thick coating" (Mo?). Also, explain the requirements for the maximum roughness value for this type of coating. It is also recommended to present a cross section of a sample showing the Mo and MoO3 coatings produced.
- After line 137, a diagram of the superimposed or created layers should be presented.
- Fig. 3 In the description below the picture, explain the abbreviations Ref. 1 and Ref. 2.
- Line198 Were the load parameters different during the run-in than in the further test? What happened to the wear products, were they removed after lapping or continued to affect the surfaces of the samples? Explain the issue of lapping in more detail in the text. You should also provide the friction results in the system Y = X ± δ for tests with and without the coating in a separate table. Prove that for two repetitions you are able to give reliable values. For the coated variant, you did up to 6 repetitions (which were almost 6 times different from each other), and not only 2. Why?
- Figure 4, Figure 6 and Figure 8 in the figure also show surface topography maps (roughness views)
- Line 233 "After the test, it was noted that the surfaces of the bearing washers show a clear topographical and mechanical change after a test duration of 38 min and that the operating life under these conditions had been reached." What was considered as the criterion for reaching the end-of-life? Provide the appropriate source or standard. From Fig. 3, seizure does not appear. From Fig. 4, no pitting is formed. The value of 45 Nm for the reference variant is achieved after about 10-20 minutes, not after 38.
- Line 242 "The operating life is the time until the cut-off criterion (max. Frictional torque of 45Nm) is reached." Why did you consider 45 Nm as the cut-off criterion? Why did you not use this criterion in the tests without Mo coating, after all, there you found the durability at the level of 38 minutes, and from Fig. 5 it follows that the criterion of 45 Nm breaks through in less than 20 minutes?
- Fig 5 is not clear. In Fig. 5a, three groups of plots can be seen which pierce the 45 Nm level from the bottom at significantly different times (500 min, 1000 min and 2900 minutes). The waveforms should be marked with colors and described with symbols.
- Line 277 Explain why did you consider coatings to prevent fretting? Rather, fretting occurs in nominally stationary nodes, only there are micro-movements between mating surfaces. In addition, a special test is used for fretting tests in the presence of grease and there is a separate standard. Why didn't you use it in your research.
- Figures 4c and 6c Both figures show other values ​​for Ra and Rz for "Bearing washer in initial state with measured roughness parameters Ra and Rz". Explain why these values ​​differed by about 10%: Ra = 0.122 for Fig. 4 (uncoated variant) and Ra = 0.135 for Fig. 6 (coated variant). Estimate what effect this had on the results of your friction measurements.
- Line 306 It would be advantageous to present a chemical composition study deep into the steel surface (GDOES). Have such studies been carried out?
- Line344 Explain why you write about the effects of research on fretting (see my earlier comment on fretting), rather you should mention tribo-oxidation.
- Line 349 did not show an increase in the durability of coated versus uncoated steel surfaces, only a longer time until a frictional moment of 45 Nm occurs. During the test, there were no phenomena that prevented the operation of the bearing, e.g. pitting or seizing. Moreover, due to the different number of repetitions (2 vs. 6) of the trials, they are not comparable. Moreover, the determination of durability most often requires the determination of the L10 parameter on the basis of the Weibull test (minimum 7 trials, preferably approx. 21). The durability of the tribological system should be clearly determined. Include my remark in your conclusions.
Author Response
Dear Reviewer,
Thank you very much for the review of my paper. I have followed your comments as far as possible and formatted them as "yellow text" in the PDF/File.
I would like to comment on some of your points here again:
- See changes in Title
- See changes in Line 1 - 16
- The tribological systems have reached extreme torque (up to 50 Nm) (see Line 224). The test rig was at the limit. We also compare the operating time with unlubricated systems. The operating time is not the same as the fatigue life of bearings. The rolling bearing fatigue life is reached when corresponding pittings can be detected on the surface. I tried to declare the definition of ‘operating time’ in Line 192-195 as well as Line 265-271.
- See changes in Line 110
- The label 1.3505 was added.
- See changes in Line 139.
- Figure 1 has been modified.
- See comment 7.
- See changes in Fig 3.
- see Table 3 (new one!)
- Modified Fig 4, 6 & 8
13a: See Line 238. The test rig and the measurement equipment used came to the mechanical limit. At the same time, the damage analysis of the reference tests then showed that the surfaces of the bearing washers were heavily worn.
13b: The experiments without Mo were to find out the cut-off criterion.
The reference tests were stopped at around 50 Nm. In order to carry out further tests without damaging the test rig, a value of 45 Nm was defined (Line 268) as the cuff-off criterion for the Mo System.
14. See Table 3. I hope the information from the table will help to answer this question.
15. For example, there is a method that tests rolling bearings with grease (ASTM D4170-10). However, an oscillating movement is considered there. In this case, we are first investigating the effectiveness of the Mo system on rotating rolling bearings in general.
17. The EDX mapping should only prove that Mo is still on the surface after the test. The high concentration of oxygen within the wear track suggests that a MoOx (MoO3) is still present. We assume that a new formation of the lubricating MoO3 has occurred and that the Mo (reservoir) serves as a lubricant source but also as wear protection. This was proven by preliminary investigations using XPS. However, this is not part of the investigations.
18. See changes (yellow) in Part ‘Conclusion’.
19. Thank you for this comment. I have bravely taken your comment into account in the summary/conclusion. See changes (yellow) in Part ‘Conclusion’.
The number of samples is too small for a Weibull evaluation. You are right about that. However, the attempt was also not to make a statistically reliable statement about the fatigue life. Much more, we wanted to verify the effectiveness of the molybdenum in the rotating components.

Reviewer 2 Report
This paper presents research on the frictional torque evolution and wear of cylindrical roller bearings lubricated with solid Mo and MoO3 particles obtained from coatings deposited on the bearing races. Friction tests, SEM analysis, and EDS results confirm that coating bearing races with thin Mo and MoO3 coatings are a reliable solution for high-temperature applications.
The research is well done, but some information is missing. My review suggestions that can improve this work are as follows:
- The use of the term "fretting" is not appropriate for the research presented. Fretting means wear resulting from small-amplitude oscillations of the mechanisms, and this is not the case in this study. An appropriate word is "friction", not "fretting".
- The results of friction tests are scattered. Indicate the surface roughness values for the bearing elements tested, if possible.
- Indicate the type, location and number of acceleration sensors (see line 160).
- Why were the tests carried out at one force and one speed and why at such a low speed?
- Provide values for threshold temperature and vibration level (see line 172). How were these values determined? Why are there no results reported for the temperature and vibration level developed?
- Given the spread of results obtained for both uncoated and coated bearings, measurements of roughness and cross-sectional thickness for both Mo and MO3 coatings could explain the different bearing behavior (see Figure 5).
- Line 314: Has there been any transfer of cage material onto the rolling elements detected by EDS analysis?
- What is the nature of the particles in Figure 8b?
- Present some values of nanometer hardness and Young's modulus for the coatings tested, as in [7] and [10].
Author Response
Dear Reviewer,
Thank you very much for your comments and the review of my paper.
I have highlighted the notes within the document in yellow,
In the following, I would like to discuss the individual points again:
1 .See changes in Line 97 and the Conclusion. Fretting was the wrong word to discribe the wear reactions. This has now also been taken into account within the Conclusion. Line 378, Line 370.
2. I measeured the roughness of the presented specimen and put the informations in the pictures. See Fig. 4, 6 and 8!
3. See changes in Line 182 and in the Fig, 2
4. For wear tests in the FE8 test rig, these shaft speeds are common. At the same time, we still have very high sliding speeds (relative movements) in the Hertzian contact.
The loads have been selected in analogy to preliminary investigations. At the same time, the load collectiv is common in the priority program DFG SPP2074.
5. Unfortunately, I could not detect a limit temperature in the reference experiment. The reason for this is the very short test duration of 38 min. The molybdenum-based coating systems, on the other hand, worked significantly longer (see Fig. 5a-c). A comparison of the temperature developments was not useful.
6. To answer this point, i added a table. See Tabl. 3
7. No wear particles were found in the raceways or the wear track itself in the EDX. However, I determined that the wear of the cage was higher with the uncoated bearings. And this with simultaneous significantly shorter running duration. However, these particles were located outside the running track. Gravimetric measurements will help in the future. This point will be further focused on in the next research steps.
8. That was Mo particles. We found this information by using edx. I put this information now in the pic. See changes Fig 8b and the new one Fig 8c
9. See changes in Line 146 - 159
Many thanks again! I hope I have taken everything sufficiently into account.
Best regards
Dennis Konopka

Reviewer 3 Report
some very small remarks through the text (comments are written directly in the pdf file).
my main question is the comment with line 269... to be considered. I don't have the answer, it is mostly a 'philosophical' question about how to interpret the results.

Author Response
Dear Reviewer,
Thank you very much for your review of our paper.
I have highlighted the changes and modifications within the document in yellow.
Additionally i have answered your comments in the old file/document. Unfortunately I can only upload one file here.
About the Review:
Especially the point with Fretting I have considered within this document. You will find changes to this in the introduction as well as in the conclusion.
about your main question (Line 276 (in old version)): In pin-on-disc investigations, exactly this behavior was found. The friction behavior was particularly good in systems with a top layer (MoO3) in contrast to pure molybdenum systems.
Furthermore, we tried to deposit the MoO3 armoph on the bearing washers. Preliminary investigations showed that the particles (MoOx) can be finely dispersed in the contact. This caused a friction reduction (measured on the friction tribometer (oscillating test)).
And a quick side note: The dissolved Mo particles were identified as MoO3 particles by an EDX and also by an additional XPS analysis. Therefore, it can be assumed that the lubricant MoO3 can be reformed by the molybdenum reservoir under tribological stress.
If necessary, I will be happy to send you the commented PDF.
Attached you will find here the new version of the paper. Thank you very much and best regards
Dennis Konopka

Round 2
Reviewer 1 Report
Thank you for considering or clarifying most of my comments. I will recommend publishing the article.
Reviewer 2 Report
The authors responded to all the questions and amended the paper as requested. I recommend this paper for publication in the Coatings journal.